# Polyphenols as Potential Attenuators of Heat Stress in Poultry Production

**DOI:** 10.3390/antiox8030067

**Published:** 2019-03-18

**Authors:** Ruizhi Hu, Yujia He, Muhammed Adebayo Arowolo, Shusong Wu, Jianhua He

**Affiliations:** 1College of Animal Science and Technology, Hunan Agricultural University, Changsha 410128, China; rzhi479@163.com (R.H.); mbayor88@gmail.com (M.A.A.); 2College of Veterinary Medicine, Hunan Agricultural University, Changsha 410128, China; hyj39306@163.com

**Keywords:** heat stress, oxidative stress, polyphenols, poultry

## Abstract

Heat stress is a non-specific physiological response of the body when exposed to high ambient temperatures, which can break the balance of body redox and result in oxidative stress that affects growth performance as well as the health of poultry species. Polyphenols have attracted much attention in recent years due to their antioxidant ability and thus, can be an effective attenuator of heat stress. In this paper, the potential mechanisms underlying the inhibitory effect of polyphenols on heat stress in poultry has been reviewed to provide a reference and ideas for future studies related to polyphenols and poultry production.

## 1. Introduction

Heat stress is a stress response that can lead to various harmful impacts on livestock productivity, such as high animal morbidity, mortality, and reduction in growth performance, directly resulting in dramatic economic losses to the livestock industry. Global warming has created a massive challenge for the livestock industry, especially in tropical and subtropical zones which contribute most to global livestock production [1]. Compared with other animals, poultry is more sensitive to heat stress, which weakens their immunological functions and makes them more susceptible to infection by pathogens, leading to a decline in growth performance and in many cases, death [2,3,4]. Furthermore, heat stress may affect meat color and pH and is recognized as one of the primary influencing factors of meat quality [5,6,7]. Generally, growth performance decline caused by heat stress is directly related to a reduction in feed intake, but increasing evidence has shown that heat stress may induce reactive oxygen species (ROS) and cause anti-oxidant system disorders, which affect nutrient absorption and metabolism [8,9]. The crosstalk between heat stress and oxidant stress, as well as the effects and mechanisms of polyphenols on stress are reviewed in the following sections.

## 2. Heat Stress in Poultry

Poultry has no sweat glands, and respiratory hyperventilation is poultry’s primary mechanism of dissipating heat [10]. The optimum temperature for most poultry species is from 18 °C to 20 °C. However, when the temperature is above this range, breathing frequency will increase three-fold to overcome the heat stress [1,11]. Most poultry species will experience heat stress when the temperature rises above 32 °C, accompanied by physiological and metabolic disorders [12].

Multiple studies have shown that heat stress can significantly reduce feed intake, daily gain, and feed utilization [13]. Due to poultry’s feathers which cover their body and lack of sweat glands, a continuous high temperature will inhibit energy metabolism [10,14]. Poultry feed intake declines 1.5% for each degree rise in temperature when the temperature ranges from 21 °C to 30 °C, and the decline in feed intake will increase to 4.6% with a temperature range of 32 °C to 38 °C [15]. When poultry are exposed to heat stress, their body temperature, blood circulation, and peripheral blood flow increases sharply, whereas their visceral blood flow decreases. These changes lead to limited nutrients utilization and thus reduce the poultry’s production performance and feed conversion efficiency [16,17]. There is a significant negative correlation between feed intake and environmental temperature [15]. The effect of heat stress on feed intake in poultry is a complex process. On one hand, heat stress decreases the gastrointestinal motility and prolongs gastric emptying, which in turn results in lowered feed intake [18]. On the other hand, poultry will increase their water intake under heat stress and thus reduce the concentration of digestive enzymes in the intestinal tract, which can affect food digestion and absorption [19].

It is reported that heat stress negatively affects the laying performance and egg quantity of poultry. Exposure of laying hens to high temperatures resulted in a significant decrease in egg weight, shell weight, and shell thickness [20,21,22,23]. Due to metabolic differences and higher heat production by laying hens than broilers, heat stress has more significant impact on laying hens than on broilers. In a heat stress experiment looking at hens laying eggs, Mashaly et al. [17] reported that the weights of eggs in the heat stress group (35 °C and 15% relative humidity, 4 h/d) were significantly lower than in the control group throughout the five week experiment. Zhu et al. [22] reported that exposure to 32 °C could significantly decrease egg weight, laying rate, egg yield, and eggshell quality.

## 3. Heat Stress and Oxidative Stress

Under normal conditions, the oxidation system and antioxidant system of poultry are in a state of dynamic equilibrium. Once the body produces too much ROS or the body’s antioxidant system is damaged, the equilibrium will be broken and cause oxidative stress [24]. This phenomenon reduces the feed intake of poultry and affects the metabolism of the body [25]. Reactive oxygen species are one of the free radicals that can exist independently and contain one or more unpaired electrons [24]. Generally, ROS and reactive nitrogen species (RNS) are the primary free radicals that participate in various metabolic reactions in the body. However, some ROS are produced in the free radical reaction process and do not strictly belong to free radicals, though they can directly or indirectly trigger the free radical reaction [24]. Mitochondrial dysfunction caused by heat stress is the basis of oxidative stress. In the early stages of acute heat stress, the levels of mitochondrial substrate oxidation and electron transport chain activity increase, resulting in excessive ROS. In the later stages of acute heat stress, uncoupling proteins (UCPs) are downregulated and excessive ROS causes damage to the protein, lipid, and DNA, which reduces mitochondrial energy production efficiency and increases production of reactive oxygen species causing mitochondrial dysfunction and increasing the oxidative stress of the body. However, chronic heat stress can reduce the metabolic capacity of mitochondria due to it upregulating the UCPs, downregulating antioxidant enzymes, and depleting the body’s antioxidants reserves which causes accumulation of ROS—breaking the oxidative balance and inducing oxidative stress [26].

Reactive oxygen species include the superoxide ion (O_2_•^−^), hydrogen peroxide (H_2_O_2_), and hydroxyl radical (•OH), and the concentration of ROS is a direct marker reflecting the level of oxidative stress in the body [24]. Superoxide ion is produced enzymatically, and nicotinamide adenine dinucleotide phosphate oxidase (NADPHox) is considered as an essential enzyme to promote the production of O_2_•^−^ in macrophages and endothelial cells [27]. In a non-enzymatic way, the electron can be directly transferred to oxygen (O_2_•^−^) because external inducers interfere with enzyme activity, for example by decreasing the activity of coenzymes [24]. The superoxide ion is the precursor of most ROS, which can produce H_2_O_2_ through disambiguation [28]. Hydrogen peroxide can spread throughout the mitochondria and across the cell membrane into other cells and finally form water by glutathione peroxidase (GSH-Px) reducing glutathione (GSH) to oxidized glutathione (GSSG) [29]. However, hydrogen peroxide also reacts with transition metal ions to generate •OH, which is known as the Fenton reaction. Agarwal et al. [30] reported that heat stress could induce carbonylation of serotransferrin releasing Fe^2+^ by increasing the Fenton reaction and producing more ·OH and H_2_O_2_. High ROS level can disrupt the cell membrane and release cytochrome C (Cyto-C), which completes assembly immediately to activate the cascade reaction of cysteine, and subsequently increases the expression of aspartic acid proteolytic enzyme 3 (Caspases-3) to induce apoptosis [26,31].

Heat shock protein 70 (Hsp70) can decrease the expression of Caspases-3 and apoptosis by suppressing the expression of the apoptotic protease activator (Apaf-1) and the release of Cyto-C. Heat stress can increase the expression of Hsp70, which can reduce ROS production by increasing the activity of antioxidant enzymes such as superoxide dismutase (SOD) [32]. However, ROS can inactivate Hsp70 and aggravate heat stress [32].

Reactive nitrogen species (RNS) including nitric oxide (NO) and nitrite peroxide (ONOO–) are also a significant cause of stress. Nitric oxide is produced in cells with L-arginine acid as a substrate through catalysis by nitrous oxide synthase (NOS), which can directly show the level of oxidative stress [33]. Heat stress can upregulate the *NOS* gene and cause overproduction of NO, resulting in the activation of various cell signaling pathways [34]. Nitric oxide can react with oxyhemoglobin to produce nitrogen dioxide (NO_2_), but also can react with ROS to produce ONOO–, which can inactivate manganese superoxide dismutase (Mn-SOD) and iron superoxide dismutase (Fe-SOD) [35]. In addition, NO can react with polyunsaturated fatty acids to produce lipid peroxide (LOO–) and lipid hydroperoxide (LOOH), which can cause lipid peroxidation and the damage of protein and DNA [26].

### 3.1. Heat Stress and the Antioxidant Enzyme System

Superoxide dismutase is the most critical protective enzyme for the elimination of superoxide anion free radicals in various tissues and organs since it can transform highly reactive O_2_•^−^ to low reactive H_2_O_2_ and keep O_2_•^−^ at a certain level in the body [36]. Catalase (CAT) is an iron-containing enzyme mainly found in different tissues including red blood cells to decompose H_2_O_2_ and eliminate the toxic effect of H_2_O_2_ [37]. Glutathione peroxidase, a selenium-containing enzyme widely distributed in the body, can promote the conversion of harmful substances produced in the lipid peroxidation reaction into corresponding alcohols and block the chain cycle reaction, it can also decompose H_2_O_2_ into H_2_O [9].

Heat stress is divided into acute heat stress and chronic heat stress with the participation of different antioxidant enzymes. Under acute heat stress, ROS level in the body is rapidly increased and the antioxidant enzyme system also responds rapidly, by which the activity of CAT, SOD, and GSH-Px are increased significantly to remove excessive free radicals. Pamok et al. [38] reported that after four days of acute heat stress (38 ± 2 °C), the GSH-Px activity was increased together with serum malondialdehyde (MDA), which can reflect the degree of oxidative damage in poultry. Vesco et al. [39] also reported that acute heat stress (38 °C for 24 h) increased the expression of the *GSH-Px* gene. However, chronic heat stress can break the antioxidant enzyme system and cause the ROS accumulation in the body to induce oxidative stress by decreasing the activity of CAT, SOD, and GSH-Px. Lu et al. [40] reported that chronic heat stress (32 °C for seven days) increased the muscle ROS level and MDA content with reduced SOD and GSH-Px activity in poultry.

### 3.2. Heat Stress and the Antioxidant Non-Enzyme System

Besides the antioxidant enzyme system, there is an antioxidant non-enzyme system which includes vitamin C, vitamin E, GSH, carotenoid, and the microelements copper, zinc, selenium, and manganese [23]. These non-enzyme substances participate in biotransformation in the body. Most of the non-enzymatic substances are obtained from food intake. Under heat stress, feed intake of poultry is reduced and thus the intake of non-enzymatic substances will be decreased, which results in oxidative stress [11]. Vitamin E is an essential antioxidant in biological systems and can penetrate into the lipid bilayer structure to combine with vitamin C and other antioxidant systems to terminate lipid oxidation [41]. Heat stress may also make the body secrete excessive glucocorticoid, which can promote the body’s decomposition of proteins and thus cause cell damage. Vitamin C can regulate the secretion of the glucocorticoid and relieve the cell damage caused by heat stress [42]. Zinc is a major component of the copper–zinc superoxide dismutase (CuZn-SOD) which plays a vital role in antioxidant defense systems [42]. Glutathione peroxidase is a selenium-dependent enzyme, meaning more selenium is needed when the body is under heat stress. Lack of selenium will reduce the activity of GSH-Px [39].

## 4. Potential Mechanisms Underlying Attenuation of Oxidative Stress by Polyphenols in Heat Stressed Poultry

Phytochemicals with antioxidant activity offer great hope as a solution for heat stress in poultry. As one of the critical secondary metabolic substances, polyphenols widely exist in a variety of plants and have been used for various purposes because of their strong antioxidant ability [43]. Polyphenols are characterized by phenol as the basic skeleton, and polyhydroxy substitution of the benzene ring can be classified into phenolic acids, acetophenones, phenylacetic acid, hydroxycinnamic acids, coumarins, naphthoquinones, xanthones, stilbenes, and flavonoids [43]. The following sections summarize the potential mechanisms underlying the antioxidant functions of three common polyphenols (as shown in Figure 1) based on our previous studies and recent peer-reviewed studies.

### 4.1. Resveratrol

Resveratrol, 3,5,4’-trihydroxystilbene, is a kind of natural plant antitoxin, which belongs to the category of non-flavonoid polysaccharides and is also an antibiotic secreted by plants to resist fungal infection under pathogen attack. Resveratrol exhibits lipophilic characteristics which lead to high absorption, but the systemic bioavailability of resveratrol is relatively low. The plasmatic concentration of resveratrol plus total metabolites is around 400–500 ng/mL (≈2 μM) after a 25 mg oral dose [44]. An isotope tracer experiment has shown that radioactivity derived from ^14^C-resveratrol can be detected in various organs such as the liver, kidney, brain, heart, lung, testis, and intestine after oral intake of ^14^C-resveratrol. This suggests that resveratrol or its metabolites can enter almost all organs after intake [45]. Resveratrol shows a strong antioxidant ability in poultry. Sahin et al. [46] reported that quails supplemented with 400 mg/kg resveratrol had a lower serum MDA concentration (*p* < 0.05) and a higher serum vitamin E concentration (*p* < 0.05). Zhang et al. [47] revealed that supplementation of 400 mg/kg resveratrol increased muscle glycogen content and the activities of total superoxide dismutase (T-SOD) and GSH-Px (*p* < 0.05) but decreased muscle MDA content and lactate dehydrogenase (LDH) activity (*p* < 0.05) in transport stress-impaired broilers. Liu et al. [48] also reported that supplementation of 200, 400, or 600 mg/kg resveratrol efficiently attenuated heat stress in black-boned chickens, and the dose of 400 mg/kg showed the strongest antioxidant effect.

Many studies have reported that resveratrol can effectively scavenge ROS, regulate the activity of various antioxidant enzymes, reduce DNA damage, and significantly enhance the expression levels of various antioxidant enzymes and proteins. The antioxidant effect of resveratrol is stronger than that of vitamin C, and it is more effective in scavenging ·OH. Das et al. [49] reported that dietary supplementation of resveratrol can inhibit lipid peroxidation and improve enzyme (SOD, GSH-Px, CAT) activity (*p* < 0.05) in hepatocytes, thus relieving liver damage caused by heat stress in rats. Zhang et al. [50] also reported that resveratrol could protect against the heat stress-impaired meat quality of broilers by increasing the muscle total antioxidant capacity (T-AOC) and activity of antioxidant enzymes (CAT, GSH-PX). Our previous study revealed that resveratrol could significantly increase the average daily feed intake (ADFI) and average daily weight gain (ADG) of black-boned chickens and reduce feed to gain ratio (F/G ratio) under the condition of heat stress. This occurred as a result of the restoration of antioxidant enzyme activity that was decreased by heat stress [48,51]. Liu et al. [32] reported that resveratrol could alleviate intestinal injuries by increasing mRNA and protein expression of Hsp70, Hsp90, and NF-kappa B, and suppressing the production of epidermal growth factor (EGF) in the mucosa.

### 4.2. Curcumin

Curcumin (feruloyl methane) is a yellow polyphenol extracted from the traditional Chinese medicine, zingiber plant. Since the discovery of curcumin in 1815, it has become one of the most widely used natural pigments. Animals can easily absorb curcumin. Ravindranath et al. [52] reported that about 60% was absorbed after oral administration of 400 mg curcumin to rats. In another rat experiment conducted by Marczylo and colleagues [53], curcumin was found in the plasma (16.1 ng/mL), urine (2.0 ng/mL), intestinal mucosa (1.4 mg/g), liver (3671.8 ng/g), kidney (206.8 ng/g), and heart (807.6 ng/g) after oral intake of 340 mg/kg of curcumin. Curcumin can therefore effectively distribute to all parts of the body after intake, potentially undergo metabolic O-conjugation to curcumin glucuronide and curcumin sulfate, as well as bio-reduction to tetrahydrocurcumin, hexahydrocurcumin, octahydrocurcumin, and hexahy-drocurcuminol [54].

The 1,1-diphenyl-2-picryl-hydrazyl free radical (DPPH) and 2,2′-azino-bis (3-ethylbenzthiazoline -6-sulfonic acid) (ABTS) radical scavenging ability of curcumin is higher than that of artificial antioxidant butylated hydroxyanisole (BHA) [55]. Many studies have shown that curcumin can improve poultry’s growth performance under heat stress [56,57,58]. Curcumin can restore the impaired growth performance caused by heat stress potentially due to its capacity to mitigate the broilers’ mitochondrial dysfunction and enhance the mitochondrial biogenesis caused by heat stress. Zhang et al. [58] reported that curcumin decreased the ROS production of broilers after heat stress by increasing the mitochondrial Mn-SOD activity and gene expression of thioredoxin 2 and peroxiredoxin-3. Curcumin can also decrease the mitochondrial malondialdehyde levels and increase mitochondrial glutathione content as well as the activities of GSH-Px, glutathione S-transferase (GSST), and Mn-SOD [57].

### 4.3. Epigallocatechin Gallatel

Epigallocatechin gallate (EGCG) is the primary component of green tea extract, which possesses strong antioxidant and anti-inflammatory properties and shows higher bioavailability than other polyphenols. The absolute bioavailability of EGCG in mice is 26.5% according to Lambert et al. [59]. Epigallocatechin gallate undergoes methylation, glucuronidation, and sulfation in vivo [59], but mostly presents in free form in the plasma and can thus effectively spread around the body [60]. Suganuma et al. [61] directly administered the [H-3] (−)-epigallocatechin gallate ([H-3]EGCG) into the mouse stomach and found significant radioactivity in the digestive tract, liver, lung, pancreas, mammary gland, skin, brain, kidney, uterus ovary, and testes. Multiple studies have reported that EGCG can attenuate heat stress in poultry. Xue et al. [62] revealed that supplementation of 300 and 600 mg/kg EGCG in heat-stressed broilers could increase growth performance in a dose-dependent manner. Luo et al. [63] also reported that a supplement of 600 mg/kg EGCG in heat-stressed broilers showed the best antioxidant property. Sahin et al. [64] reported that supplement with 400 mg/kg EGCG in heat-stressed quails showed the best antioxidant property. Thus 400–600 mg/kg of EGCG may be the optimal dosage in poultry.

Supplemental EGCG can maintain the equilibrium of oxidation–reduction and improve the expression of antioxidant genes (*SOD*, *CAT*, *GSH-Px*), thus reducing the damage caused by oxidative stress [65]. Supplemental EGCG can improve the poultry’s growth performance and alleviate the oxidant damage under heat stress by modulating the antioxidant enzymes [62,63,64]. Epigallocatechin gallate enhances antioxidant enzyme activity mainly due to the fact that it can activate nuclear factor (erythroid-derived 2)-like 2 (Nrf2) pathways. Sahin et al. [64] reported that EGCG improves antioxidant capacity by regulating the transcription factor Nrf2 as well as Nrf2-regulated heme oxygenase -1(HO-1). However, Orhan et al. [66] reported that EGCG could also decrease hepatic expression of activator protein-1 (AP-1), cyclooxygenase-2 (COX-2), and heat shock proteins (Hsps). Thus, EGCG potentially attenuates heat stress through modulating stress-responsive transcription factors.

The beneficial effects of polyphenols as a potential attenuator of heat stress are mainly attributed to their antioxidant activities. Polyphenols can act as radical scavengers depending on their chemical structures [67] and thus scavenge ROS including O_2_•^−^, •OH, and H_2_O_2_ to eliminate the oxidative damage caused by heat stress [68]. The phenol functional group of polyphenols can donate a hydrogen atom to the free radicals and direct quenching ROS. Also, they may block the action of some enzymes (e.g. xanthine oxidase and protein kinase C) that directly generate O_2_·^−^ [69,70]. Furthermore, polyphenols can modulate cell signaling pathways to alleviate the impact of heat stress. On the one hand, resveratrol [32], EGCG [64], and curcumin [54] have been proven as potent inhibitors of nuclear factor kappa B(NF-κB), which can transcribe inflammatory markers such as interleukin-6 (IL-6), interleukin-2 (IL-2), and tumor necrosis factor α (TNF-α) to cause inflammation [71]. On the other hand, resveratrol [72], EGCG [64], and curcumin [56] may upregulate the antioxidant response pathways such as the transcription factor Nrf2 mediated antioxidant enzymes to improve the antioxidant enzyme system. Under a heat stress situation, resveratrol and EGCG can enhance the activity of antioxidant enzymes (CAT, GSH-Px, SOD) [50,51] and non-enzyme systems (GSH [51], vitamin E [46]), while curcumin can increase the activity of Mn-SOD, GSH-Px, and GSST [56]. In addition, polyphenols may attenuate heat stress by regulating the expression of heat shock proteins. Resveratrol can upregulate the transcription level of Hsp70 and Hsp90 in the immune organs of black-boned chickens under heat stress and directly participate in the immune response against the damage caused by heat stress [48]. However, EGCG has been reported to suppress the elevated hepatic expression of Hsps caused by heat stress, specifically inhibiting the expression of Hsp70 and Hsp90 by restraining the promoter activity [66]. To our knowledge, there is no direct evidence to demonstrate the mechanism of polyphenols alleviating damage caused by heat stress through the inhibition of Hsps expression.

The bioavailability of polyphenols remains controversial due to the low absorption rate since polyphenols undergo various metabolisms such as methylation, glucuronidation, and sulfation after intake [59]. Generally, the concentration of the parent structure of polyphenols rather than their metabolites were detected in early studies [73]. However, recent studies have shown that phenolic metabolites possess strong biological activities, thus the bioavailability of polyphenols should not exclude their phenolic metabolites [74]. The stomach and small intestine are the main absorption sites of polyphenols. An isotope tracer experiment with resveratrol showed that the concentration of radioactivity in the intestinal tract was the highest [45]. The concentration of curcumin in intestinal mucosa can reach 1.4 mg/g after intake of 340 mg/kg curcumin [53]. Around 33% of EGCG can be excreted in feces [60]. Therefore, the gut should be the main place for the function of polyphenols. The liver is the main metabolic organ and also a target of polyphenols. The concentration of curcumin in the liver can reach 3.6 mg/g at a dose of 340 mg/kg [53]. High concentrations of radioactivity were detected in the liver in the isotope tracer experiment with resveratrol and EGCG [45,60]. The high concentration of polyphenols in the liver can potentially explain the hepatoprotection ability under heat stress. In addition, polyphenols can also reach circulation and other organs. Plasma concentration of curcumin can reach 16.1 ng/mL at a dose of 340 mg/kg [53]. Radioactivity can be detected in the kidney, testis, lung, pancreas, skin, mammary gland, and even brain after intake of isotope labeled resveratrol and EGCG [44,61]. This may also explain why polyphenols are capable of attenuating systemic disorders caused by heat stress.

## 5. Conclusions

Taken together, heat stress is a significant cause of economic loss in poultry production and is almost inevitable. It mainly leads to a decrease in growth and reproductive performance, and impacts the meat quality of poultry by inducing oxidative stress in the body. Improving the antioxidant capacity of poultry may help mitigate the influence of heat stress. Polyphenols are natural antioxidants that can reduce oxidative stress and widely exist in plants, they therefore have great potential to be used as a novel feed additive for improving productivity in heat-stressed poultry.

## Figures and Tables

**Figure 1 antioxidants-08-00067-f001:**
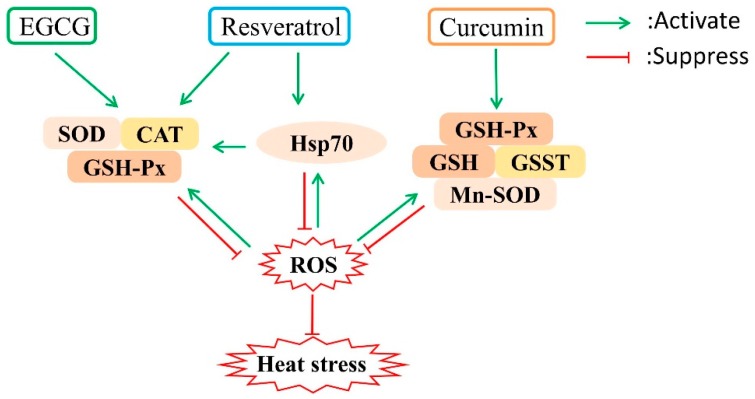
Potential mechanisms underlying the protective effect of polyphenols against heat stress. Polyphenols can upregulate the expression of stress response proteins such as heat shock proteins and antioxidant enzymes, which can suppress reactive oxygen species (ROS). EGCG, epigallocatechin gallate; SOD, superoxide dismutase; CAT, catalase; GSH-Px, glutathione peroxidase; Hsp70, heat shock proteins 70; GSH, glutathione; GSST, glutathione S-transferase; Mn-SOD, manganese superoxide dismutase.

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
