# Peer review of "Polyphenols as Potential Attenuators of Heat Stress in Poultry Production"

_antioxidants, 2019, doi:10.3390/antiox8030067_

Reviewer 1 Report

The aim of the present paper is to review existing evidence about possible roles of polyphenols, which can reduce heat stress, major problem in poultry production, often occurring especially in countries with hot climate. Birds under heat stress are prone to immunosuppression, weight loss, decrease in eggs production and also increased mortality is observed. The losses in flock are dependent on the birds age, frequency, time of exposure to the elevated temperature and humidity in the environment. Here, important protective measures are presented to minimize heat stress and to prevent health problems during production cycle in poultry. Animal feed provides a range of antioxidants that help the body building an integrated antioxidant system responsible for a prevention of damaging effects of free radicals and products of their metabolism. Polyphenols extracted from plants have multiple functions in animal production. They are natural antioxidants that can reduce oxidative stress, and widely exist in plants, thus have great potential to be used as a novel feed additive for improving productivity in heat-stressed poultry. In my opinion, the material in this study is very valuable and should not to be ignored, just ought to be continued, especially in the age of global warming.

Author Response

We appreciate the positive comments from the reviewer. Yes, heat stress is one of the major problems in poultry production, and that is why we wrote this review. 

Reviewer 2 Report

The subject of the manuscript is very current and of great impact to scientific world and  public opinion, therefore this work is very useful.

However, I believe that some minor changes are needed:

Figure 1 is ambiguous: the authors should better explain in the legend the different meaning of the arrows cointinue (green) and dashed (red).

If the green arrows indicate "activation", it could be understood that the polyphenols activate the antioxidants which, in turn, activate the ROS, which induce oxidative stress. The understanding of the figure should be made less ambiguous.

Other minor revisions:

page 2, line 90: reducing glutathione (GSSG) to oxidized glutathione (GSH)

probably it's a typo: the reduced glutathione is GSH, the oxidized glutathione is GSSG, the opposite of what is indicated in the text.

Page 2, line 91

However, H2O2 reacts with transition-metal ions generated to · OH, which is known as Fenton.

Complete the sentence with Fenton reaction

Page 3, line 98

..... release of to-c.

What does to-c mean? maybe cyt c or what? it is necessary to elucidated

Author Response

Figure 1 is ambiguous: the authors should better explain in the legend the different meaning of the arrows cointinue (green) and dashed (red).

Response: Thank you for the valuable suggestions. Yes, we have revised the figure and added explanation in the captions. Thank you.

page 2, line 90: reducing glutathione (GSSG) to oxidized glutathione (GSH)

Response: Thank you for the careful reviewing. We have corrected this mistake in the revised manuscript.

Page 2, line 91

However, H2O2 reacts with transition-metal ions generated to · OH, which is known as Fenton.

Complete the sentence with Fenton reaction

Response: We are sorry for the ambiguity. Yes, we have completed the sentence with Fenton reaction in the revised manuscript.                                                                       

Page 3, line 98

..... release of to-c.

What does to-c mean? maybe cyt c or what? it is necessary to elucidated

Response: We are sorry for the typos. It has been corrected to “Cyto-C” in the revised manuscript. Thank you.